# A reference DNA barcode library for Austrian amphibians and reptiles

**Lukas Zangl**[1,2☯*], **Daniel Daill**[1,3‡], **Silke Schweiger**[4‡], **Georg Gassner**[4‡], **Stephan Koblmüller**[1☯*]

**1** Institute of Biology, University of Graz, Graz, Austria, **2** Studienzentrum Naturkunde, Universalmuseum Joanneum, Graz, Austria, **3** Consultants in Aquatic Ecology and Engineering—blattfisch e.U., Wels, Austria, **4** First Zoological Department, Herpetological Collection, Museum of Natural History Vienna, Vienna, Austria

☯ These authors contributed equally to this work.
‡ These authors also contributed equally to this work.
* lukas.zangl@uni-graz.at (LZ); stephan.koblmueller@uni-graz.at (SK)

**Data Availability Statement:** The data underlying this study can be found at BOLD (www.boldsystems.org) using the link: dx.doi.org/10.5883/DS-BCAHF. In addition, data have been uploaded to GenBank and are available using the accession numbers MN993072 - MN993264.

## Abstract

In the last few years, DNA barcoding became an established method for species identification in biodiversity inventories and monitoring studies. Such studies depend on the access to a comprehensive reference data base, covering all relevant taxa. Here we present a comprehensive DNA barcode inventory of all amphibian and reptile species native to Austria, except for the putatively extinct *Vipera ursinii rakosiensis* and *Lissotriton helveticus*, which has been only recently reported for the very western edge of Austria. A total of 194 DNA barcodes were generated in the framework of the Austrian Barcode of Life (ABOL) initiative. Species identification via DNA barcodes was successful for most species, except for the hybridogenetic species complex of water frogs (*Pelophylax* spp.) and the crested newts (*Triturus* spp.), in areas of sympatry. However, DNA barcoding also proved powerful in detecting deep conspecific lineages, e.g. within *Natrix natrix* or the wall lizard (*Podarcis muralis*), resulting in more than one Barcode Index Number (BIN) per species. Moreover, DNA barcodes revealed the presence of *Natrix helvetica*, which has been elevated to species level only recently, and genetic signatures of the Italian water frog *Pelophylax bergeri* in Western Austria for the first time. Comparison to previously published DNA barcoding data of European amphibians and reptiles corroborated the results of the Austrian data but also revealed certain peculiarities, underlining the particular strengths and in the case of the genus *Pelophylax* also the limitations of DNA barcoding. Consequently, DNA barcoding is not only powerful for species identification of all life stages of most Austrian amphibian and reptile species, but also for the detection of new species, the monitoring of gene flow or the presence of alien populations and/or species. Thus, DNA barcoding and the data generated in this study may serve both scientific and national or even transnational conservation purposes.

**Funding:** Financial support was provided by the Austrian Federal Ministry of Science, Research and Economy in the frame of the ABOL (Austrian Barcode of Life; www.abol.ac.at) pilot project on vertebrates and an ABOL associated project within the framework of the "Hochschulraum-Strukturmittel" Funds. One of the authors (Daniel Daill) is employed by a commercial company: Consultants in Aquatic Ecology and Engineering, Austria. We note that he contributed most of his work during the time of his Masters' thesis at the University of Graz and got employed by this company only recently. This company provided support in form of salary for him, but did not have any additional role in the study design, data collection and analysis, decision to publish, or preparation of the manuscript. The specific roles of all authors are articulated in the 'author contributions' section.

**Competing interests:** The authors declare that there is no competing interest. Daniel Daills commercial affiliation does not alter our adherence to all PLoS One policies on sharing data and materials.

## Introduction

Amphibians and reptiles -at least across Europe- comprise rather species poor taxonomic groups compared to other classes of vertebrates [1]. However, despite their low species diversity, they are important indicators for biomonitoring and conservation management due to their sensitivity to environmental changes [1–7]. They show high vulnerability to changes in water regime, land use, pollution, habitat disruption, fragmentation and destruction and changes in interspecific competition accompanied by novel pathogens, like the chytrid fungus infesting amphibians or *Ophidiomyces ophiodiicola*, the snake fungal disease [4,5,8,9,10,11]. All of these factors have led to a decline in population and species numbers, not only on a local, but also on a global scale [5,12–16]. Generating and maintaining a comprehensive picture of the status of threatened species and in order to promote conservation efforts [12,17–21], environmental monitoring is an undisputable necessity. Furthermore, as a member of the EU, Austria -as anchored in the EU Habitats Directive- is obliged to frequently report on the status of protected species and habitats [22]. Assessments of species composition, distribution and welfare frequently requires the species level identification of amphibians and reptiles in the field. While determination of adult or fully-grown specimens is usually a routine exercise for experts, identification of leftovers from road kills, eggs or larvae of closely related species such as newts (*Triturus* spp.) and frogs (*Rana* spp., *Pelophylax* spp.) can pose a bigger challenge [7]. These challenges may often be overcome by DNA barcoding, a method that compares short, standardized gene sequences with a reference database [23]. This method has been shown to yield high accuracy and success rates for species identification, although certain exceptions and limitations remain [7]. Thus, in the recent past a large number of DNA barcoding sequences of various amphibian and reptile species accumulated across the globe [1,7,24–28]. All of this data contributes to the global iBOL initiative [29] and can be used either for direct comparison or as the basis for environmental DNA (eDNA) approaches for studies on biodiversity, population dynamics, range shifts and anthropogenic translocation of species [7,30,31].

For Austria 20 species of amphibians and 14 species of reptiles are currently recognized, although *Vipera ursinii* is considered to be extinct [32,33]. Despite sporadic findings of palmate newts (*Lissotriton helveticus*) in Vorarlberg in 2008 and 2009 [34], this species does not appear in national species catalogues. Furthermore, the recently described *Natrix helvetica*, which was elevated to species level from *N. n. helvetica* in 2017 [35], can also be found in Austria, but is not yet listed as a distinct native faunal element. However, both *L. helveticus* and *N. helvetica* will be incorporated in the next red list of Austrian amphibians and reptiles (S. Schweiger, unpubl. data). Special conservational concern is attributed to a total of 76% or 16 species of amphibians and 10 species of reptiles, as they are mentioned in the appendices II and IV of the EU Habitats directive and all of these species are furthermore subject to national conservation laws as well [22]. The Austrian Barcode of Life initiative (ABOL, www.abol.ac.at) aims at contributing to the global genetic species inventory as well as providing a comprehensive overview of the national herpetofauna. With this data release we provide 194 DNA barcodes of all species native to Austria, except for the putatively extinct in Austria *V. ursinii rakosiensis* and the rarely encountered *L. helveticus*. In addition, we discuss the genetic diversity of Austria's herpetofauna in a European context by comparing it to previously published molecular data from amphibians and reptiles from surrounding countries.

## Material and methods

Most samples were obtained from natural history museums. Additional samples were collected in the field, resorting only to freshly dead specimens to avoid sacrificing live animals (permit

numbers ABT13-53S-7/1996-156 and ABT13-53W-50/2018-2, or passed on by the Museum of Natural History in Vienna, a CITES-registered federal institution, which is allowed to receive and store samples in its collections). Overall, 239 samples of Austrian amphibians and reptiles were obtained. Species of the water frog (*Pelophylax*) complex were determined morphologically using [36], crested newt species (*Triturus*) identification followed [37] and was linked to the geographic region the samples were acquired from. Tissue samples were stored in pure ethanol in a freezer at -20˚C, and reptile and amphibian voucher specimens were fixed in 70% and 50% ethanol respectively and permanently stored in natural history museums. All information regarding specimen, collection and storage is available on BOLD (www.boldsystems.org, project code 'BCAHF') (also see S1 Table). DNA extraction followed two methods. As standard method, we employed a rapid Chelex protocol [38]. In addition, some difficult samples were extracted with the DNeasy Blood & Tissue Kit (Quiagen), following the manufacturer's instructions. Polymerase chain reaction (PCR) for the first part of the mitochondrial cytochrome oxidase subunit 1 gene (COI), gel electrophoresis, enzymatic clean-up using Exo-SAP-IT and chain termination sequencing followed [39] and [40]. Primers used for PCR and cycle sequencing are listed in S2 Table, annealing temperatures ranged from 46–50˚C. DNA fragments were purified with SephadexTM G-50 (Amersham Biosciences) and visualized on an ABI 3130xl capillary sequencer (Applied Biosystems). The sequences were edited manually and an alignment was manually created and trimmed in MEGA 6.06 [41]. For further analysis, the alignment was split into a reptilian and an amphibian dataset. Neighbor-joining (NJ) trees based on the Kimura 2-parameter (K2P) [42] distance model were generated on BOLD using the Taxon ID tree analysis tool for visualization of taxonomic clades. To put the genetic diversity of the Austrian herpetofauna into a European context, a second set of NJ trees was calculated with MEGA 6.06 after including sequences from other European countries [7,26,28,43–47], downloaded from the online repositories GenBank and BOLD. Maximum intraspecific genetic distances as well as the minimum interspecific distances were calculated under the K2P model using the "Barcode Gap Analysis" tool implemented on BOLD [48].

## Results

Out of 239 samples, we generated 194 DNA barcodes with a length of 584 to 658 bp, conforming to an 81% sequencing success rate. All sequences were deposited on GenBank (Accession Nos. MN993072—MN993264) and BOLD (dx.doi.org/10.5883/DS-BCAHF). Barcodes were obtained for all amphibian and reptile species native to Austria, except for the presumably extinct *V. ursinii rakosiensis* and the only rarely reported *L. helveticus*. Furthermore, *COI* sequence data proved the presence of *N. helvetica*, which has been elevated to species level only recently, and the first ever recovered genetic signatures of the Italian water frog (*P. bergeri*) in Austria. Overall, DNA barcodes of Austrian amphibians and reptiles contributed to 31 already existing BINs on BOLD and created seven new BINs (*Iberolacerta horvathi*, *Natrix natrix*, *Zootoca vivipara* (3), *Vipera ammodytes* and *Pelophylax* kl. *esculentus*). Most of the species (28 out of 34) were represented only by a single BIN and no cases of BIN sharing were detected, except for newts and water frogs, where hybrids are possible. In cases where species are represented by two or more BINs, this is due to single deviating sequences (*Rana temporaria*), or distinct intraspecific lineages (*Podarcis muralis*, *Zootoca vivipara*). Separate NJ trees for amphibians (Fig 1) and reptiles (Fig 2) were generated based on the COI sequences, allowing for unambiguous identification of all species except for the hybridogenic species complex of water frogs (*Pelophylax* spp.) and the crested newt complex (*Triturus* spp.), for which morphological species assignment was not reflected perfectly by the NJ tree.

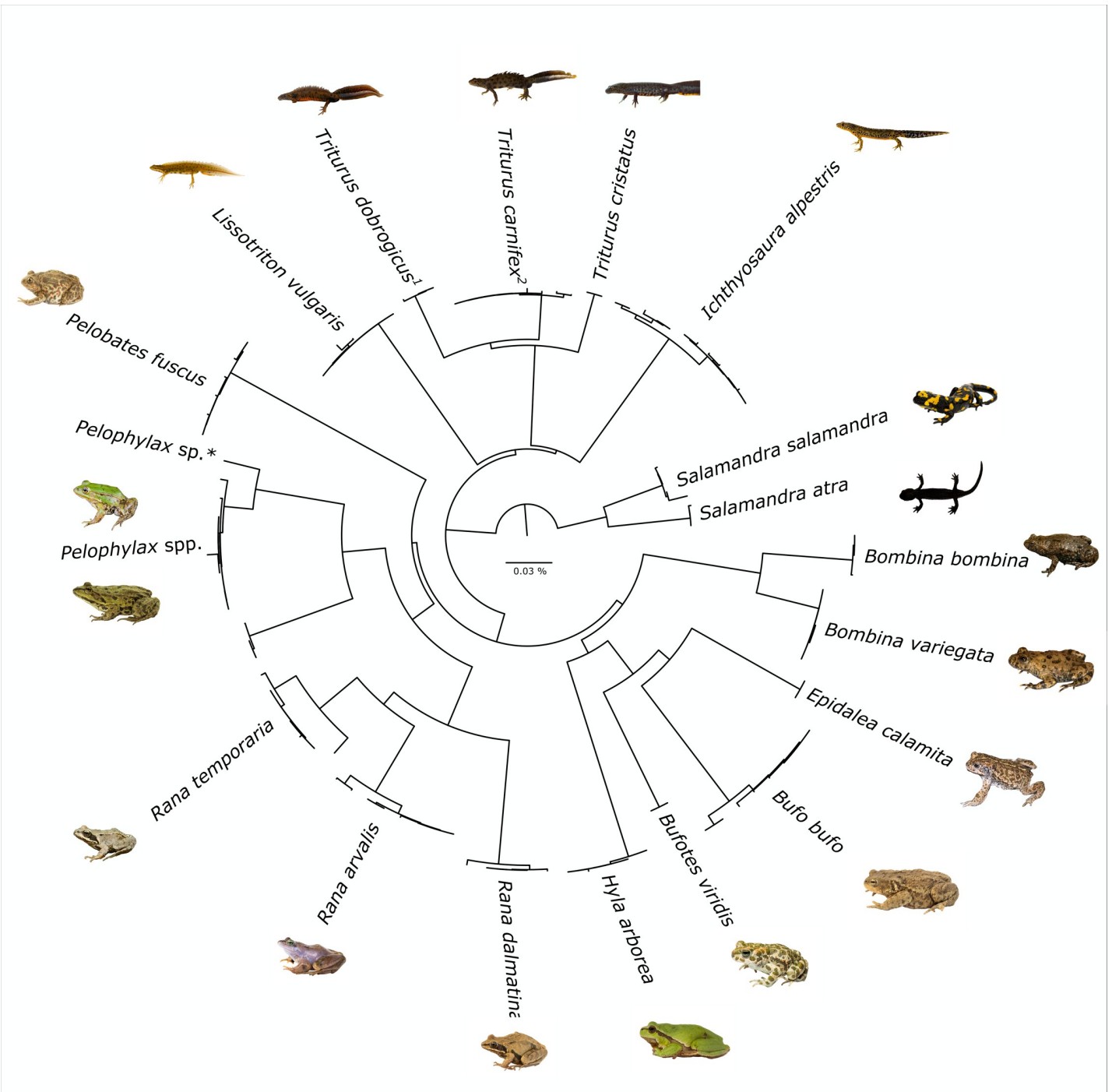

**Fig 1. NJ tree of Austrian amphibians based on K2P distances.** The tree was inferred with the "Taxon ID Tree" tool implemented in BOLD and visualized in FigTree v1.4.2 (http://tree.bio.ed.ac.uk/software/figtree/). * indicates an Austrian water frog sample showing genetic signatures of the Italian water frog (*P. bergeri*). [1] indicates the *T. dobrogicus* clade, which contains one sample identified as *T. carnifex*. [2] marks the *T. carnifex* clade, which also holds one *T. cristatus*.

Austrian *Pelophylax*, determined based on their morphology, formed three clades of one (mitochondrial DNA of *P. bergeri*), eight (2 *P. esculentus*, 5 *Pelophylax* sp. and 1 *P. lessonae*) and three (2 *P. ridibundus* and 1 *P. lessonae*) sequences, respectively. The comparison of

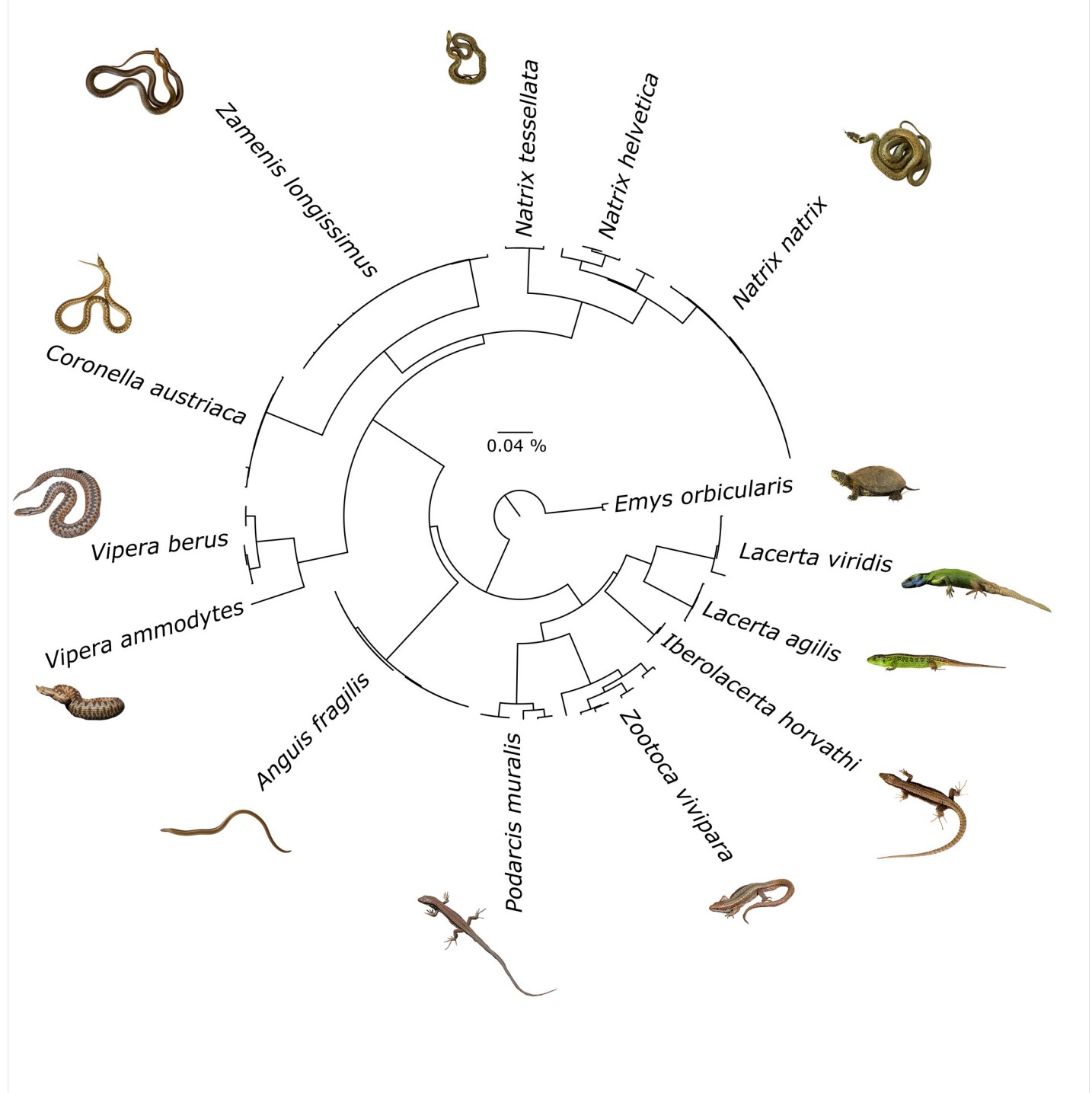

**Fig 2. NJ tree of Austrian reptiles based on K2P distances.** The tree was inferred with the "Taxon ID Tree" tool implemented in BOLD and visualized in FigTree v1.4.2 (http://tree.bio.ed.ac.uk/software/figtree/).

Austrian water frog *COI* sequences to already existing European water frog data corroborate this result (Fig 3). Although one clade (BOLD:AAN3045) was mainly composed of *P. ridibundus* sequences from Germany and Austria, it also included one German and one Austrian *P.*

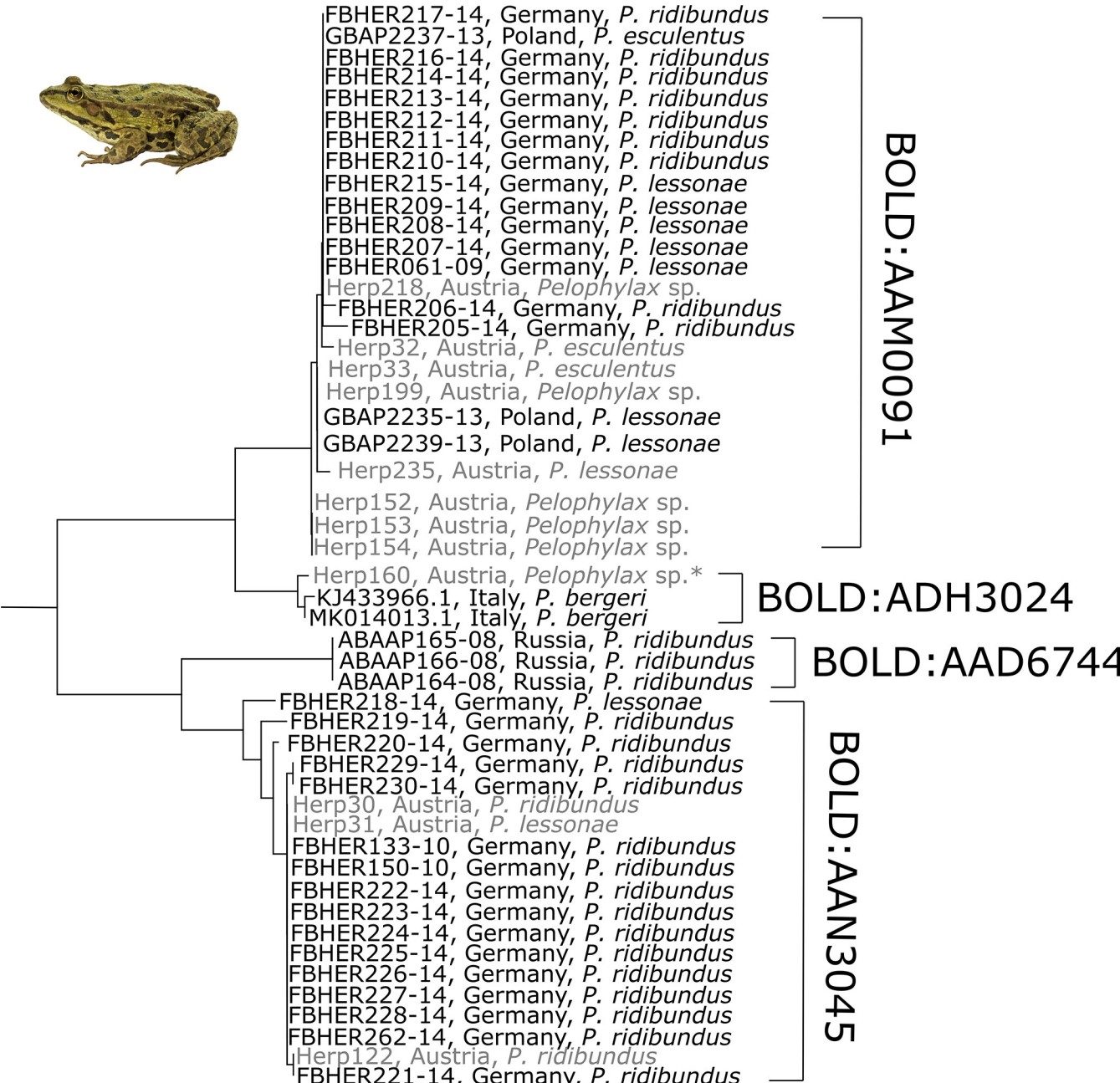

**Fig 3. Subtree of the hybridogenetic *Pelophylax* species complex.** DNA barcodes do not clearly resolve the two parental (*P. lessonae*, *P. ridibundus*) and the hybrid (*P.* kl. *esculentus*) species but provide evidence for a genetic signature of the Italian water frog (*P. bergeri*) in one Austrian water frog sample (marked with *). Light gray labels represent samples processed in this study.

*lessonae* sample. *P. ridibundus* from Russia formed a distinct clade and was also represented by an individual BIN (BOLD:AAD6744). Two sequences of Italian *P. bergeri* clustered together with one Western Austrian water frog sample in a distinct BIN (BOLD:ADH3024), indicating the presence of *P. bergeri* mtDNA in (western) Austrian water frogs. The second major clade of pan-European water frog *COI* sequences included a mixture of all three species occurring in

Central Europe. *P.* kl. *esculentus*, *P. lessonae* and *P. ridibundus* from Austria, Germany and Poland all contributed to the same BIN (BOLD:AAM0091).

Intraspecific genetic distances were well below 2% for most of the species (Table 1). In some species, haplotypes were very similar and differed by only a few substitutions despite a fairly large number of samples included in the analyses (e.g., *Zamenis longissimus*, 13 samples, 0.49% maximum intraspecific distance ($I_{max}$); *Anguis fragilis*, 14 samples, 0.46% $I_{max}$). Higher intraspecific genetic distances were observed for *N. natrix* (5.94%), *V. berus* (2.65%) *P. muralis* (3.6%), *Z. vivipara* (6.09%) within the reptiles and *R. temporaria* (4.01%), *Triturus carnifex* (8.00%) and *Triturus cristatus* (7.84%) within the amphibians (Table 1).

The large intraspecific distances in newts, however, are due to hybridization/introgression in areas of sympatry. Calculation of within (*T. carnifex* 0%, *T. cristatus* 0.03%, *T. dobrogicus* 0.02%) and between BIN distances (*T. carnifex* 7.5%, *T. cristatus* 7.8%, *T. dobrogicus* 7.5%), however, perfectly comply with the barcode gap hypothesis. Minimum interspecific distances within reptiles (10.76%) and amphibians (9.92%) exceeded intraspecific distances considerably. Contrasting the generally observed pattern of low intra- and higher interspecific genetic distances, the species complex of water frogs (*Pelophylax* spp.) showed an exactly reversed pattern when analyzed based on presumed species assignment. When resorting to BINs, genetic distances of conspecifics did not exceed one percent and the interspecific distance between BOLD:ADH3024 (new BIN, this study) and BOLD:AAM0091 (*P. lessonae* according to [7] amounted to 3.85%.

## Discussion

### General barcoding success and efficiency

In this study we present 194 barcodes for all extant species of the Austrian herpetofauna, except for the only recently documented and rarely observed palmate newt (*L. helveticus*) and the putatively extinct in Austria meadow viper *V. ursinii rakosiensis*. For all species, two or more barcodes were generated, except for the nose-horned viper (*V. ammodytes*), for which only a single sample could be obtained. Analysis of genetic barcoding data almost perfectly reflects the country's species assemblage. Of the 11 families, 22 genera and 34 species of amphibians and reptiles occurring in Austria, only the hybridogenetic species group of water frogs (*Pelophylax* spp.) and the crested newt species (*Triturus* spp.), for which hybrids are known to exist in areas where two or more species occur in sympatry [49–51], could not be resolved properly by the *COI* tree. This result was also reflected by the barcode gap analysis. Consequently, a species level determination based on DNA barcoding is possible for all species of the Austrian herpetofauna that form distinct barcode clusters. This, in principle, also includes the crested newts, but with the caveat that potential hybrids cannot be detected based on *COI* data alone.

### The problem with *Pelophylax*

The only exception where reliable species identification was not possible with DNA barcodes is the genus *Pelophylax*. Even though the three species might be distinguished based on morphological and bioacoustical characters, mtDNA does not allow for species identification because of cross-breeding between the hybridogenetic *P.* kl. *esculentus* with its parent species [7]. This circumstance is shown by our taxon ID tree of Austrian *Pelophylax COI* sequences (Fig 1), as well as by the tree including also other European water frog *COI* data (Fig 3). Obviously, morphologically determined specimens of all three species do contribute to the same BIN on BOLD, thus blurring the significance of certain BINs. This also implies, that neither a BLAST search on GenBank or BOLD, nor the accumulation of further *COI* data of these

**Table 1. Genetic (K2P) distances (in %) within and between species.**

| Species | BIN | N | $I_{max}$ | Nearest neighbor | DNN |
|---|---|---|---|---|---|
| Anura | | | | | |
| Bombinatoridae | | | | | |
| *Bombina bombina* | BOLD:AAD1964 | 3 | 0.15 | *Bombina variegata* | 10.09 |
| *Bombina variegata* | BOLD:AAD4416 | 5 | 0.17 | *Bombina bombina* | 10.09 |
| Bufonidae | | | | | |
| *Bufo bufo* | BOLD:AAC2139 | 8 | 1.41 | *Epidalea calamita* | 18.02 |
| *Epidalea calamita* | BOLD:AAI8496 | 2 | 0 | *Bufotes viridis* | 17.44 |
| *Bufotes viridis* | BOLD:AAJ8500 | 2 | 0 | *Epidalea calamita* | 17.44 |
| Hylidae | | | | | |
| *Hyla arborea* | BOLD:AAN9979 | 5 | 0.3 | *Bufo bufo* | 24.29 |
| Pelobatidae | | | | | |
| *Pelobates fuscus* | BOLD:AAL6663 | 6 | 0.46 | *Rana temporaria* | 25.75 |
| Ranidae | | | | | |
| *Pelophylax* spp. | BOLD:ADH3024 | 1 | 0* | BOLD:AAM0091 | 3.85 |
| | BOLD:AAM0091 | 8 | 0.96* | BOLD:ADH3024 | 3.85 |
| | BOLD:AAN3045 | 3 | 0.15* | BOLD:ADH3024 | 13.11 |
| *Rana arvalis* | BOLD:AAL1420 | 8 | 1.54 | *Rana temporaria* | 9.92 |
| *Rana dalmatina* | BOLD:AAM0090 | 5 | 0.76 | *Rana temporaria* | 14.25 |
| *Rana temporaria* | BOLD:AAL6095 | 6 | 4.01 | *Rana arvalis* | 9.92 |
| | BOLD:ACH4056 | 1 | | | |
| Caudata | | | | | |
| Salamandridae | | | | | |
| *Ichthyosaura alpestris* | BOLD:AAC5105 | 12 | 1.85 | *Triturus carnifex* | 19.74 |
| *Lissotriton vulgaris* | BOLD:AAL6213 | 7 | 0.66 | *Ichthyosaura alpestris* | 20.98 |
| *Salamandra atra* | BOLD:ACM1022 | 3 | 0.15 | *Salamandra salamandra* | 9.49 |
| *Salamandra salamandra* | BOLD:ACE6170 | 5 | 1.54 | *Salamandra atra* | 9.49 |
| *Triturus carnifex* | BOLD:ACE8564 | 9 | 8* | *Triturus dobrogicus* | 0.15 |
| *Triturus cristatus* | BOLD:AAC3031 | 2 | 7.84* | *Triturus carnifex* | 0.15 |
| *Triturus dobrogicus* | BOLD:AAE0668 | 3 | 0* | *Triturus carnifex* | 0.15 |
| Squamata | | | | | |
| Anguidae | | | | | |
| *Anguis fragilis* | BOLD:AAK0900 | 14 | 0.46 | *Iberolacerta horvathi* | 24.73 |
| Colubridae | | | | | |
| *Coronella austriaca* | BOLD:AAL9606 | 7 | 0.64 | *Zamenis longissimus* | 12.75 |
| *Zamenis longissimus* | BOLD:AAL5946 | 13 | 0.49 | *Coronella austriaca* | 12.75 |
| *Natrix natrix* | BOLD:AAL6710 | 12 | 5.94* | *Natrix tessellata* | 10.76 |
| | BOLD:ACM1720 | 2 | | | |
| | BOLD:AAX3380 | 3 | | | |
| | BOLD:ADH1094 | 1 | | | |
| *Natrix tessellata* | BOLD:AAN4201 | 3 | 0.47 | *Natrix natrix* | 10.76 |
| Viperidae | | | | | |
| *Vipera ammodytes* | BOLD:ADH3451 | 1 | 0.00 | *Vipera berus* | 12.00 |
| *Vipera berus* | BOLD:AAW7158 | 2 | 2.65 | *Vipera ammodytes* | 12.00 |
| | BOLD:ACM2231 | 3 | | | |
| Lacertidae | | | | | |
| *Iberolacerta horvathi* | BOLD:ADG8839 | 3 | 0.15 | *Lacerta agilis* | 16.70 |
| *Lacerta agilis* | BOLD:AAL6669 | 4 | 0.31 | *Lacerta viridis* | 13.60 |

*(Continued)*

**Table 1.**  (Continued)

| Species | BIN | N | $I_{max}$ | Nearest neighbor | DNN |
|---|---|---|---|---|---|
| *Lacerta viridis* | BOLD:AAJ3146 | 5 | 1.88 | *Lacerta agilis* | 13.60 |
| *Podarcis muralis* | BOLD:AAH9270 | 3 | 3.60 | *Zootoca vivipara* | 17.16 |
| | BOLD:AAL6640 | 3 | | | |
| *Zootoca vivipara* | BOLD:ADH1152 | 1 | 6.09 | *Podarcis muralis* | 17.16 |
| | BOLD:ADH1309 | 3 | | | |
| | BOLD:AAL6569 | 2 | | | |
| | BOLD:ADH1153 | 2 | | | |
| Testudines | | | | | |
| Emydidae | | | | | |
| *Emys orbicularis* | BOLD:AAF8183 | 2 | 0.92 | *Lacerta viridis* | 25.15 |

K2P distances of *COI* sequences within and between species studied. BIN, "Barcode Index Number" assigned by BOLD; N, number of barcode sequences contributing to a certain BIN; $I_{max}$, maximum intraspecific distance; Nearest neighbor, most closely related species; DNN, genetic distance to the closest related species.

* Indicates ambiguous cases where hybridization or multiple species blur genetic distances.

species will result in an unambiguous identification, unless this data is supported and verified by additional analyses (e.g. PCR-RFLP [52]; microsatellite data [53–54]; PCR—sequence length differences [55]). On the other hand, DNA barcoding revealed the mitochondrial signature of the Italian water frog *P. bergeri* in one of our samples (Fig 3) collected in Vorarlberg in the far West of Austria for the first time. This again highlights one of the strengths of DNA barcoding, as it can be used to detect human-induced translocations or track natural migrations triggered by climate change, both possibly leading to a turnover in local species assemblages [56–58].

## Signatures of postglacial recolonization

Most other species of both amphibians and reptiles are characterized by low intra- and higher interspecific genetic distances and represented by single BINs, which is in line with the findings of [7]. However, there are some exceptions like *Bufo bufo* (1.41% $I_{max}$), *Rana arvalis* (1.54% $I_{max}$), *Salamandra salamandra* (1.54% $I_{max}$), *Ichthyosaura alpestris* (1.85% $I_{max}$) or *Lacerta viridis* (1.88% $I_{max}$), where genetic distances are slightly higher but still contribute to only a single BIN, and others like *V. berus* (distance within clusters 2.65%), where the split into

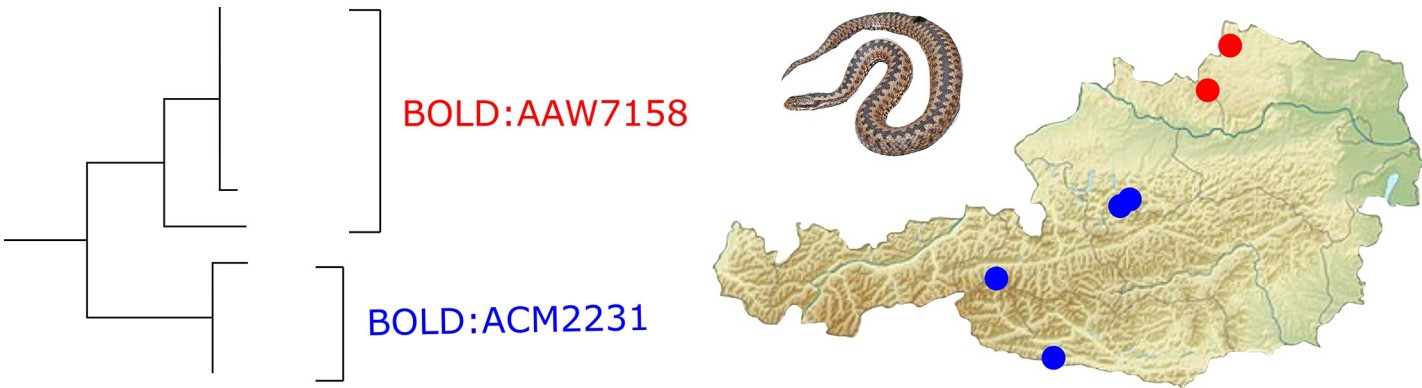

**Fig 4. Subtree of the common European adder.** *COI* sequences of *V. berus* allow for an assignment of origin from an inner alpine (blue) or adjacent lowland regions (red). Clades also include sequences from [7], only Austrian samples are displayed on the map.

two separate lineages is also reflected by separate BINs. Similar to [7], divergent lineages in *V. berus* were also detected in Austria, allowing for a clear assignment of individuals to either an inner alpine area or to adjacent lowland regions (Fig 4). These lineages might be explained by different glacial refugia and post-glacial recolonization routes [59–60].

The genetic sub-structuring observed in the *COI* data of European populations of common toad (*B. bufo*), moor frog (*R. arvalis*), fire salamander (*S. salmandra*) and alpine newt (*I. alpestris*) has been attributed to different glacial refugial areas and explains the increased intraspecific genetic distance observed in Austrian samples [61–64]. In contrast to *S. salamandra* (Fig 5A), where our data is perfectly in line with [7], the alpine newt data generated in the present study deviates from [7] in that there is a clear separation into two conspecific lineages, comprising samples from north and from south(east) of the Alps, respectively (Fig 5B). This is in line with [64], who suggested two separate Pleistocene refugia north and south of the Alps. However, all 12 Austrian samples of *I. alpestris* are included in the same BIN (BOLD: AAC5105) and cluster together with samples from Germany, Spain and the Ukraine, congruent with *cytochrome b (cytb)* and *16S* data [64].

In the case of *Rana temporaria*, one divergent haplotype is causing the large intraspecific distance of 4.01%, which is also reflected by a new BIN (BOLD:ACH4056). Unlike the rest of the Austrian *R. temporaria* samples, which were obtained from the inner or southern region of the Austrian Alps, this specific sample was obtained from north of the Danube in Lower Austria. In comparison with other European common frog data (Fig 5D), this particular haplotype clusters together with samples from Sweden, Russia, the Ukraine and Germany. However, although a basic separation into an Eastern and Western lineage of *R. temporaria* across mainland Europe was suggested by [66], we cautiously refrain from assigning divergent haplotypes/ BINs to one of these lineages.

Similar to *V. berus*, *Z. vivipara* shows deep conspecific lineages in the *COI* topology (Fig 2). Since they share similar ecological niches and inhabit the same habitats and biogeographical regions, it is not surprising that they also share similar postglacial recolonization patterns [58,60]. Nuclear and mitochondrial sequence data suggest up to 13 subclades and six main lineages with two areas of overlap, one being situated in Northern Italy, Austria and Northern Slovenia [58]. This could explain the high intraspecific divergence and consequently the split into four distinct BINs observed in Austrian *Z. vivipara COI* data (Fig 5F).

## Signatures of sympatric hybridization and introduction

Ambiguous results were obtained for the crested newt species (*Triturus* spp.) [67]. Genetic distances within and between BINs perfectly fit the barcode gap hypothesis and clearly separate sequence clusters and species. Based on morphological species assignment, though, the maximum within-clade distance exceeds the distance to the nearest neighbor by far (Table 1). This results from hybridization/introgression in areas of sympatry, which has been frequently reported for these three species [49–51,68]. The wall lizard (*P. muralis*) is known to occur throughout Europe in more than 100 populations originating from eight geographically distinct genetic lineages [65]. Furthermore, repeated introductions of allochthonous populations within and outside its native distribution range increased the overall distribution range and led to hybridization events between autochthonous and allochthonous subspecies [65,69–70]. Throughout Austria, at least three subspecies are known to occur, two of which, *P. muralis muralis* and *P. muralis maculiventris* are autochthonous. Thus, finding different haplotypes was expected [70]. Similar to [7], DNA barcodes clustered in more than one clade and contributed to more than one BIN (Fig 5C). However, only one BIN (BOLD:AAL6640) was shared between German and Austrian samples, the rest of the Austrian samples was represented by a

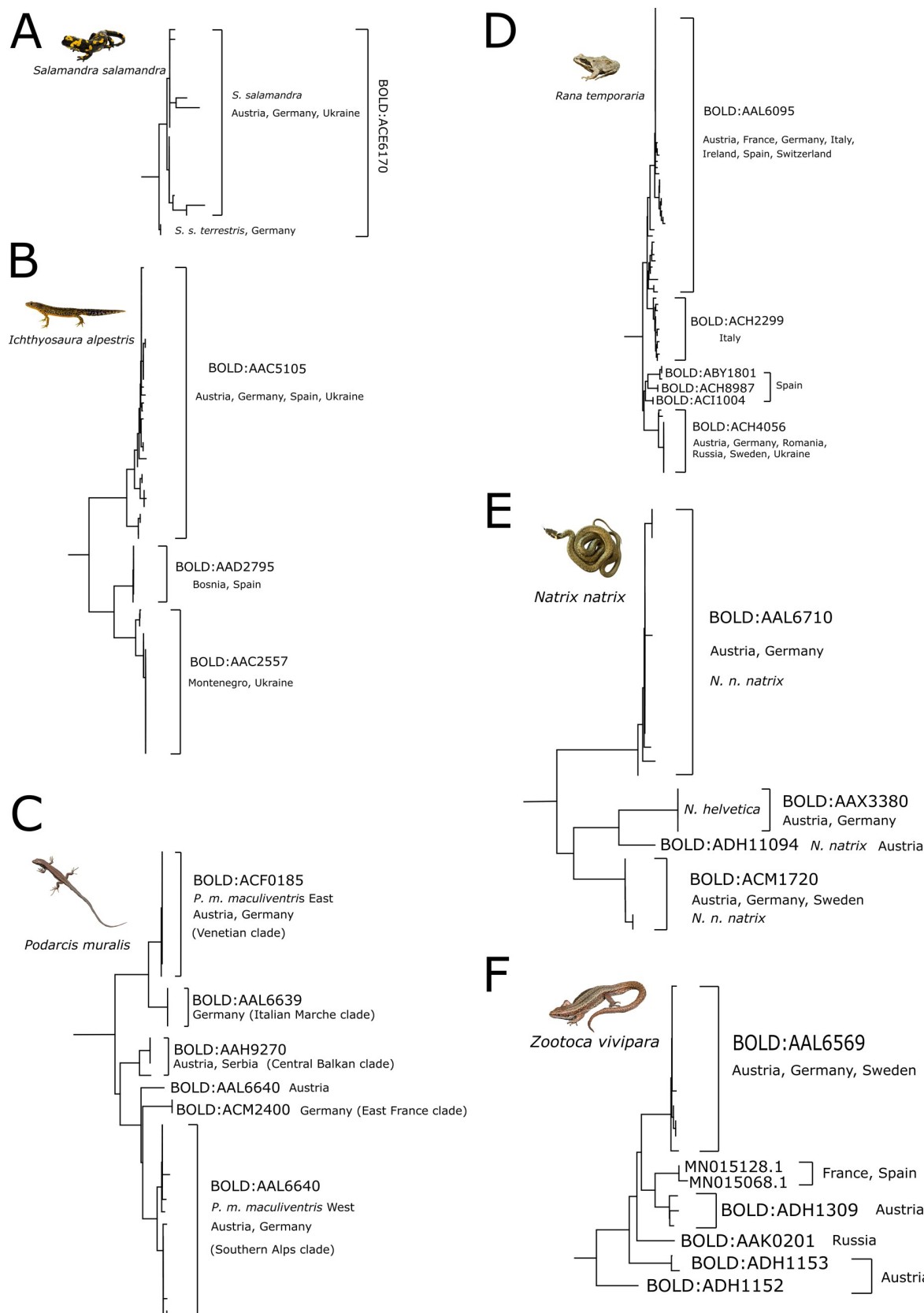

**Fig 5. Cases of ambiguity.** A) The two recognized subspecies of *S. salamandra* form two distinct clades but contribute to the same BIN. B) *I. alpestris* is represented by only one BIN despite increased intraspecific distance and a subdivision into separate clades. C) Presumed subspecies of *P. muralis*. Austrian samples are found in the three clades recovered by [7], but also in the "Central Balkan clade" proposed by [65] and another Austria specific BIN. D) Conspecific lineages within *R. temporaria* represented by two distinct BINs. E) Discrepancies within *N. natrix*. Two separate BINs were recovered within the nominal form *N. n. natrix*, as well as one new BIN from Austria. *N. helvetica* forms a distinct clade with an individual BIN. F) High genetic diversity within *Z. vivipara*. The separate clades and distinct BINs likely correspond to various genetic lineages proposed by [58].

different BIN (BOLD:AAH9270), which might represent the Central Balkan clade according to [65]. This genetic lineage likely represents the subspecies *P. m. muralis*, which should be present across large parts of the species' Austrian distribution range. Whether or not this distinct BIN is of autochthonous or allochthonous origin cannot be resolved and is outside the scope of the present study. Nonetheless, these genetic signatures may provide a geographic traceability and thus be of interest for conservation purposes.

## Systematic and taxonomic implications

Similar to [7], our *COI* data also seems to pinpoint certain taxonomic cases where subspecies might be revealed. Systematic relationships of the genus *Natrix* have been repeatedly reevaluated, leading to the erection of new genera, the elevation of former subspecies to species level and the redefinition of distribution areas, but still rendering the exact number of subspecies and their validity open to debate [35,71]. Since 2017, however, *N. helvetica* is considered a valid species with its core distribution situated west of the river Rhine [35]. DNA barcoding data generated in this study as well as the definitive morphological determination of one of the samples, though, suggests that *N. helvetica* can also be found in Austria's westernmost state Vorarlberg, as three of our samples clearly cluster with one *N. n. helvetica* from Germany (Fig 5E). Furthermore, three other distinct BINs were recovered, two within the nominal form *N. n. natrix* [72], possibly corresponding to the two main genetic lineages in Central Europe ("yellow" and "red") found by [35]. Genetic distances between BINs ranged from 2.4 to 5.3%. However, we refrain from assigning BINs to certain subspecies, as [35] already found discrepancies between distribution ranges, genetics and morphology and highlighted the necessity of a comprehensive taxonomic revision.

## Summary

In summary, DNA barcoding is a powerful tool for the identification of almost all species of amphibians and reptiles native to Austria. The only exceptions remaining are the species complex of water frogs (*Pelophylax* spp.) and syntopic hybrids of the crested newts (*Triturus* sp.), for which *COI* barcodes do not provide species level resolution. Furthermore, the species level identification of tissue remains, eggs and larval stages but also non-invasive sampling (cheek swaps, eDNA) will be possible based on a comprehensive DNA barcode reference library [7]. National -like this one- and large-scale data sets will also allow the determination of geographic origin to some extent [7]. In this respect, Austria has proven an important geographic area where various genetic lineages of several species from different refugial areas abut and overlap, and thus valuable for the understanding of the distribution of European amphibians and reptiles. However, DNA barcoding also proved valuable in the detection of new/introduced/ potentially invasive species (*N. helvetica*, *P. bergeri*) and subspecies (*N. natrix*) and can pinpoint possible allochthonous haplotypes (*P. muralis*). Thus, DNA barcoding data can also serve conservation purposes in terms of monitoring native fauna and the early detection of human mediated introduced species/populations or natural (including potentially climate change induced) immigrations.

## Supporting information

**S1 Table. Table containing sampling and storage information.** All necessary sampling and storage data as well as BOLD and BIN numbers are listed for all samples obtained and barcoded in the present study.
(XLS)

**S2 Table. List of all primers used in the present study.**
(DOCX)

## Acknowledgments

We are very grateful to Werner Kammel, Werner Stangl, Frank Weihmann, Gernot Kunz, Johanna Gunczy, Christoph Hahn and Birgit Rotter and all staff of the Österreichische Bundesforste AG in Vienna and Lower Austria for their help collecting various samples. Furthermore, we like to thank Stefan Weigl from the Biozentrum Linz and Robert Lindner from the Haus der Natur in Salzburg as well as Georg Friebe and Christine Tschisner from INATURA Dornbirn for providing samples from their museum collections and Frank E. Zachos from the Natural History Museum in Vienna for countless coordinative efforts and revision of the manuscript. We kindly acknowledge Wolfgang Gessl (www.pisces.at) and Christoph Riegler for providing the pictures of Austrian amphibians and reptiles.

## Author Contributions

**Conceptualization:** Lukas Zangl, Silke Schweiger, Georg Gassner, Stephan Koblmüller.

**Data curation:** Lukas Zangl, Daniel Daill, Silke Schweiger, Georg Gassner, Stephan Koblmüller.

**Formal analysis:** Silke Schweiger, Georg Gassner, Stephan Koblmüller.

**Funding acquisition:** Stephan Koblmüller.

**Investigation:** Stephan Koblmüller.

**Methodology:** Lukas Zangl, Daniel Daill, Stephan Koblmüller.

**Project administration:** Lukas Zangl, Silke Schweiger, Stephan Koblmüller.

**Supervision:** Silke Schweiger, Stephan Koblmüller.

**Validation:** Georg Gassner, Stephan Koblmüller.

**Visualization:** Lukas Zangl.

**Writing – original draft:** Lukas Zangl, Daniel Daill.

**Writing – review & editing:** Lukas Zangl, Daniel Daill, Silke Schweiger, Georg Gassner, Stephan Koblmüller.

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
