## [Decision Letter · Decision Letter 0]

27 Dec 2019

PONE-D-19-31052

A reference DNA barcode library for Austrian amphibians and reptiles

PLOS ONE

Dear Mr Zangl,

Thank you for submitting your manuscript to PLOS ONE. After careful consideration, we feel that it has merit but does not fully meet PLOS ONE’s publication criteria as it currently stands. Therefore, we invite you to submit a revised version of the manuscript that addresses the points raised during the review process.

Your paper was read by one referee and myself. We both consider your paper as a valuable and important contribution to European / Austrian herpetology.

The referee has pointed out some critics including the question about morphological ID of species studied (especially relevant in green frogs) among others; please see below.

We would appreciate receiving your revised manuscript by Feb 10 2020 11:59PM. To enhance the reproducibility of your results, we recommend that if applicable you deposit your laboratory protocols in protocols.io, where a protocol can be assigned its own identifier (DOI) such that it can be cited independently in the future. For instructions see: http://journals.plos.org/plosone/s/submission-guidelines#loc-laboratory-protocols

We look forward to receiving your revised manuscript.

Kind regards,

Stefan Lötters

Academic Editor

PLOS ONE

Journal Requirements:

'Financial support was provided by the Austrian Federal Ministry of Science, Research and Economy in the frame of the ABOL (Austrian Barcode of Life; www.abol.ac.at) pilot project on vertebrates and an ABOL associated project within the framework of the “Hochschulraum-Strukturmittel” Funds. '

We note that one or more of the authors are employed by a commercial company: Consultants in Aquatic Ecology and Engineering, Austria.

Reviewers' comments:

Reviewer's Responses to Questions

**Comments to the Author**

1. Is the manuscript technically sound, and do the data support the conclusions?

Reviewer #1: Yes

2. Has the statistical analysis been performed appropriately and rigorously? 

Reviewer #1: Yes

3. Have the authors made all data underlying the findings in their manuscript fully available?

Reviewer #1: Yes

4. Is the manuscript presented in an intelligible fashion and written in standard English?

Reviewer #1: Yes

5. Review Comments to the Author

Reviewer #1: The authors present a data release on the DNA barcoding campaign of Austrian reptiles and amphibians. The manuscript follows a standard structure of similar data releases and, by this, fits neatly into the existing body of literature. Consequently, the focus of the manuscript is on publishing the barcoding data. The overall presentation, writing, and language are good. I just have some minor questions and comments that should be clarified prior to publication.

Kind regards

Oliver Hawlitschek

General comments:

- How was morphological identification of species accomplished? This is particularly crucial for Pelophylax and Triturus. Please list the references / methods (and possibly the identifiers) by which / who the specimens were identified.

- Entire text: check writing: "DNA barcodes" vs. "DNA-barcodes"

- Is this the first publication of P. bergeri mtDNA in Austrian green frog samples? If yes, this should be highlighted more. If no, please cite the appropriate references.

Specific:

l. 65: Change "biodiversity assessments" to "studies on biodiversity". This facilitates understanding the entire sentence.

l. 68: Delete "singule".

l. 81: Change "V. ursinii rakosiensisi" to "V. ursinii rakosiensis". Change order: "the in Austria putatively extinct V. ursinii rakosiensisi" to "the putatively extinct in Austria V. ursinii rakosiensis".

l. 119: Change "L. helvetica" to "L. helveticus".

l. 142: Remove period / full stop "." after "BOLD".

l. 145 onwards: "Austrian Pelophylax formed three clades…" Were species names assigned based on morphological identification of specimens or on DNA barcodes? I assume this is morphological ID, but please specify.

l. 168: "Imax": Explain the abbreviation.

l. 173, Table 1: Change "Natrix tesselate" to "Natrix tesselata".

l. 197: See comment to l. 81.

l. 338: Change "personal" to "staff".

Check reference list for correct format. E.g., l. 356: change "2003; 12: 5–12." to "2003;12: 5–12."

6. PLOS authors have the option to publish the peer review history of their article (what does this mean?). If published, this will include your full peer review and any attached files.

Reviewer #1: Yes: Oliver Hawlitschek

---

## [Author Response · Author response to Decision Letter 0]

3 Feb 2020

Response to Reviewers

Journal Requirements:

1. We ensured that our manuscript meets the PLOS One style requirements, including file naming.

2a. We provided an amended Funding Statement declaring the commercial affiliation of one of the co-authors at the online forum.

2b. We provided an updated Competing Interest Statement at the online forum.

2c. We included both an updated Funding Statement and Competing Interest Statement in our cover letter and uploaded it onto the online forum. 

General comments:

- How was morphological identification of species accomplished? This is particularly crucial for Pelophylax and Triturus. Please list the references / methods (and possibly the identifiers) by which / who the specimens were identified.

Reply: We listed the references used for determination of specimens of Pelophylax and Triturus and noted the link to geography, which aids in species assignment (as hybridization zones and distribution of ‘pure’ species are already known.

- Entire text: check writing: "DNA barcodes" vs. "DNA-barcodes"

Reply: We changed it to “DNA barcodes” throughout the whole manuscript.

- Is this the first publication of P. bergeri mtDNA in Austrian green frog samples? If yes, this should be highlighted more. If no, please cite the appropriate references.

Reply: Indeed, this is the first ever record of genetic signatures of P. bergeri recovered from Austria. At the reviewers’ suggestion, we emphasized this fact by adding “…the Italian water frog Pelophylax bergeri in Western Austria for the first time.” in the abstract and the discussion and “…the first ever recovered genetic signatures…” in the results section. However, since we found it only in a single individual and since we cannot be sure whether this animal could have been transported or introduced by humans or migrated to Austria naturally, we opted to present this discovery with caution and reservation. 

Specific:

l. 65: Change "biodiversity assessments" to "studies on biodiversity". This facilitates understanding the entire sentence.

Reply: We agree and changed it to “studies on biodiversity”.

l. 68: Delete "singule".

Reply: Done.

l. 81: Change "V. ursinii rakosiensisi" to "V. ursinii rakosiensis". Change order: "the in Austria putatively extinct V. ursinii rakosiensisi" to "the putatively extinct in Austria V. ursinii rakosiensis".

Reply: Done.

l. 119: Change "L. helvetica" to "L. helveticus".

Reply: Done.

l. 142: Remove period / full stop "." after "BOLD".

Reply: Done.

l. 145 onwards: "Austrian Pelophylax formed three clades…" Were species names assigned based on morphological identification of specimens or on DNA barcodes? I assume this is morphological ID, but please specify.

Reply: Specimens were first up determined based on their morphological appearance, following best practice guidelines for the generation of DNA-barcodes. To clarify, we added “…, determined based on their morphology,…”.

l. 168: "Imax": Explain the abbreviation.

Reply: Done. We included “maximum intraspecific distance (Imax)” as an explanation for the abbreviation at first mentioning.

l. 173, Table 1: Change "Natrix tesselate" to "Natrix tesselata".

Reply: Done.

l. 197: See comment to l. 81.

Reply: Done like for line 81.

l. 338: Change "personal" to "staff".

Reply: Done.

Check reference list for correct format. E.g., l. 356: change "2003; 12: 5–12." to "2003;12: 5–12."

Reply: Done. We changed line 356 and checked the rest of the references and made the necessary corrections if needed.

---

## [Editor Report · Decision Letter 1]

5 Feb 2020

A reference DNA barcode library for Austrian amphibians and reptiles

PONE-D-19-31052R1

Dear Dr. Zangl,

We are pleased to inform you that your manuscript has been judged scientifically suitable for publication and will be formally accepted for publication once it complies with all outstanding technical requirements.

With kind regards,

Stefan Lötters

Academic Editor

PLOS ONE
---

## [Editor Report · Acceptance letter]

28 Feb 2020

PONE-D-19-31052R1 

A reference DNA barcode library for Austrian amphibians and reptiles 

Dear Dr. Zangl:

I am pleased to inform you that your manuscript has been deemed suitable for publication in PLOS ONE. Congratulations! Your manuscript is now with our production department. 

With kind regards,

on behalf of

Prof. Dr. Stefan Lötters 

Academic Editor

PLOS ONE